# MIXTURE OF QUANTIZED EXPERTS (MOQE): COMPLEMENTARY EFFECT OF LOW-BIT QUANTIZATION AND ROBUSTNESS

## ABSTRACT

Large Mixture of Experts (MoE) models could achieve state-of-the-art quality on various language tasks, including machine translation task, thanks to the efficient model scaling capability with *expert parallelism* (Fedus et al., 2021). However, it has brought a fundamental issue of larger memory consumption at deployment time. Furthermore, this results in significant inference speed degradation at auto-regressive decoding steps due to the increased memory transfers. In this paper, we propose *Mixture of Quantized Experts (MoQE)* which is a simple *weight-only* quantization method applying ultra low-bit down to 2-bit quantizations only to expert weights for mitigating the increased memory and latency issues of MoE models. We show that low-bit quantization together with the MoE architecture delivers a reliable model performance while reducing the memory size significantly even without any additional training. Especially, `expert` layers in MoE models are much more robust to the quantization than conventional feedforward networks (FFN) layers. In our comprehensive analysis, we show that MoE models with 2-bit and 80% sparse expert weights can deliver better model performance than the dense model trained on the same dataset. We present how quantization of different parts of models affects the performance with various experiments using a large MoE model (5.3 B). As a result of low-bit quantization, we show the model size can be reduced by 79.6% of the original half precision floating point (fp16) MoE model. This cuts down the model size of 5.3B parameters from 8.4x of the dense model to only 1.7x of the dense model after 2-bit quantization. It still preserves 1.88% higher accuracy than the dense model. Combined with an optimized GPU runtime implementation, it also achieves 2.7X speed-up which is even slightly faster than the FLOPs equivalent dense model.

## 1 INTRODUCTION

Large Language Models (LLMs) have shown their effectiveness on various language tasks by increasing the number of trainable parameters together with the framework of pre-training a model on a large scale data and using it to different downstream tasks (Devlin et al., 2018; Radford et al., 2018; Liu et al., 2019; Raffel et al., 2020). With the advancement of distributed large scale training methods (Shazeer et al., 2018; Rasley et al., 2020; Ren et al., 2021; Baines et al., 2021) and large scale data collection (Raffel et al., 2020; Hoffmann et al., 2022), the models get even larger and break state-of-the-art performance with the increased model capacity (Brown et al., 2020; Rae et al., 2021; Zoph et al., 2022; Zhang et al., 2022; Smith et al., 2022; Chowdhery et al., 2022). However, the cost of training these models increases whenever more parameters are added, and this may not be sustainable.

As a solution to address this issue, sparsely activated models (Shazeer et al., 2017) are more widely adopted and show significant efficiency improvements in terms of model size scaling while enabling up to trillions of parameters to be trained more efficiently and achieving better model accuracy (Lepikhin et al., 2020; Fedus et al., 2021; Kim et al., 2021; Artetxe et al., 2021). Mixture-of-Experts (MoE) models are one type of sparsely activated models replacing a single layer in a model with a group of parallel layers which are called `experts` combined with a gate layer. For a given input, the gate layer selects a subset of the `experts` from the group, and use them for processing

the input. By limiting the number of subset layers for a given input to one or two, the theoretical FLOPs stays almost constant even if we add hundreds of parallel layers into the MoE group. Thus far, most studies have shown that it is effective to increase the capacity of the models by replacing feedforward networks (FFN) of Transformer (Vaswani et al., 2017) blocks with MoE layer consists of multiple FFN layers together with a gating network (Lepikhin et al., 2020; Fedus et al., 2021; Kim et al., 2021; Artetxe et al., 2021). One of the most unique and critical components of MoE models is the gating network which decides how to conditionally select experts for each input, and there have been various studies to improve it to achieve a better training convergence ((Lewis et al., 2021; Roller et al., 2021; Zuo et al., 2021; Clark et al., 2022; Liu et al., 2022; Zhou et al., 2022) and they are well surveyed in Fedus et al. (2022).

In spite of the progress on the training of MoE models, there have been only a few handfuls of studies related to MoE model inference. Rajbhandari et al. (2022) designs a more efficient MoE architecture and distributed runtime to achieve 7.3X inference speed-up. Kudugunta et al. (2021) uses task specific information to reduce the size of the model at deployment time by only loading task specific experts. Kim et al. (2021) prunes some experts at deployment time to reduce the model size by trading-off model performance. Zoph et al. (2022) uses knowledge distillation technique to distill a large MoE model into a smaller dense model to reduce the memory consumption and improve the throughput. Even with all the proposed techniques, there has not been a solution to accelerate the inference of MoE models while maintaining the accuracy.

Quantization is a type of model acceleration and compression techniques by estimating a floating point number into a smaller precision number. There are various studies that show quantization is effective to accelerate neural network model inference (Rodriguez et al., 2018; Stock et al., 2019; Choukroun et al., 2019; Gholami et al., 2022). Especially, it has been known to be very effective in natural language generation such as machine translation ((Kim et al., 2019; Aji & Heafield, 2020; Fan et al., 2021)) and natural language understanding (Kim & Awadalla, 2020) tasks. However, there has not been an in-depth study about how quantization works with large MoE models.

Recently, Dettmers et al. (2022); Yao et al. (2022) have studied how quantization works on large scale language models. Dettmers et al. (2022) looks at outlier features in the activations of large language models, and proposes to decompose them while performing matrix multiplications. In our quantization method, this is not needed because it is a weight-only quantization and outliers in activations cannot affect the performance. And, the weights are dequantized back to fp16 while matrix multiplication is done. This also makes our approach not require a special low-bit instructions. And, we show that this can be applied to lower bits than 8-bit for large MoE models. ZeroQuant (Yao et al., 2022) presents a series of techniques including knowledge distillation (Kim & Rush, 2016) for achieving a higher quality quantization. Our focus is to exploit the intrinsic characteristics of MoE layers based on our investigation, and we show that a simple quantization algorithm can achieve significantly higher efficiency and maintain the quality at the same time.

Our contributions in this paper are as below.

- We present extensive studies about how applying low-bit (down to 2-bits) quantization to different layers of MoE transformer models affects the model accuracy together with comparisons to the corresponding dense model with the same embedding size.

- We show that expert weights are highly robust to the quantization, therefore they can be quantized to 3-bit without additional training or calibration data and to 2-bit with Quantization Aware Training (QAT) which results in 79.6% reduction in memory size. Combined with a runtime optimization, we show that the method boosts the inference speed significantly more than 2.7X faster. We leverage the memory bounded characteristic of auto-regressive decoders, so reduced memory bottleneck improves the overall efficiency even with additional dequantization steps in our procedure. Based on the observations, we propose a new framework named *Mixture of Quantized Experts (MoQE)* which is a simple *weight-only* quantization method only applied to MoE expert weights.

- Finally, we show an emerging sparsity of more than 80% in the expert weights to be zero from 2-bit quantization. The expert weight matrices are **sparse and very low-precision at the same time**, while still outperforming the dense counterpart trained on the same dataset.

## 2 BACKGROUND - CHALLENGES OF DEPLOYING MoE MODELS

In the widely used MoE architecture, even with a constant or only sub-linearly higher theoretical FLOPs by using top-1 or top-2 gating, the increased model size with additional experts has a serious negative impact on the inference performance in various aspects.

### 2.1 INCREASED MEMORY FOOTPRINT

First of all, due to the increased model size, the model requires much more accelerator memory. With modern accelerators like GPUs, the accelerator memory size is limited. So, more accelerators are required to handle 1 model which causes communication problem described next. Also, the model takes up more memory, so the batch size is limited to be small which prevents the optimal utilization of processing cores.

### 2.2 SLOWER INFERENCE SPEED

**Increased communication overhead.** In the distributed training and inference set-up for large scale models, it is natural to use many GPUs or accelerators for a single model. The model weights can be distributed across different accelerators with various techniques (Ren et al., 2021) and expert parallelism (Fedus et al., 2021). However, in Liu et al. (2022), it is shown that the communication overhead with expert parallelism at training time could take up to more than half of the entire end-to-end time depending on the number of GPUs and clusters. This could affect inference efficiency even more severely because inference usually needs fewer FLOPs numbers than training, and communication bottleneck will stand out more. Therefore, it is desirable to use as few numbers of accelerators as possible to avoid this overhead.

**Memory bandwidth bottleneck with MoE layers.** The increase in the model size not only causes communication overhead, but also brings a significant inference speed impact on the modern processor architectures. While performing beam search decoding, the size of activation (an individual token) is relatively small and the decoding operation is memory bandwidth bounded. This means transferring model weight matrices in a memory hierarchy is a huge bottleneck. With the increased number of experts, the burden of memory transfer increases even more, and directly impacts the inference speed.

**Inference speed measurement.** Table 1 shows an actual speed difference measured with dense and MoE models on an NVIDIA's V100 GPU. Two models are encoder and decoder based on the transformer architecture (Vaswani et al., 2017), and have exactly the same model settings except for the number of experts. The speed measurements are done on the translation task from German to English using auto-regressive beam search with beam size of five. Both models are evaluated on the same `PyTorch` [1] with half-precision floating point (fp16). The MoE model uses top-1 gating which assigns only one expert for a given input token which provides the same theoretical FLOPs as the corresponding dense model (with the same embedding size). Due to the excessive memory transfer caused by the increased number of experts, the actual inference speed decreases by 60% of the original dense model's speed as shown in the table.

Table 1: Inference speed measurements and model sizes of dense and MoE models. Both models run with batch size of 24 and the throughput is measured with the number of sentences processed for one second.

| Model | Throughput (sentences/second) | Model size (*fp16*) in GB | % of MoE weights |
|---|---|---|---|
| Dense | 14.02 | 1.18 | - |
| MoE (32 experts) | 5.37 | 9.91 | **92.8 %** |
| Difference | 0.38X | 8.38X | - |

To overcome these challenges, we focus on reducing the model size utilizing quantization. Especially, increased model size and latency are mostly from the expert FFN weights which contribute

---

[1] https://github.com/pytorch/pytorch

92.8 % of all weights in this specific model setting, so the FFN weights are our main target for the optimization. With an emerged sparsity in expert weights from the low-bit quantization, we also explore a further sparsification opportunity with a simple magnitude pruning technique.

## 3 QUANTIZATION METHODS FOR MoE LAYERS

There are multiple design choices to quantize model weights. In this section, we analyze the numerical characteristics of different layers in a large MoE model, and describe the decisions we have made to most effectively quantize the MoE layers.

### 3.1 NUMERICAL DISTRIBUTION OF MODEL WEIGHTS

While quantizing matrices, it is desired not to have outliers, but to have smoothly distributed numerical values. Outliers usually skew the range to be quantized and scaling factors get too large. Figure 1 shows weight distribution box plots of linear layers in the MoE model's FFN blocks. Following the widely used practice, an MoE layer is in every other layer (Lepikhin et al., 2020; Fedus et al., 2021; Kim et al., 2021). Even number layers $\{0, 2, ...\}$ are expert FFN layers and odd number layers $\{1, 3, ...\}$ are normal dense FFN layers. First, all of them are centered around zero. However, dense FFN layers have a much larger range than MoE FFN layers. This indicates that dense FFN layers have more outliers than MoE FFN layers. This phenomenon is more prevalent in the second linear layers sometimes reaching down to $-8.0$ which is shown in Figure 1b. Figure 2 shows example histograms of an expert FFN weight and a dense FFN weight. As can be seen in Figure 2b, the example dense FFN layer suffers from outliers seriously. However, expert FFN weights in Figure 2a show smooth distribution without any major outliers. We observe the similar pattern all across different layers and different experts. In Appendix C, we additionally include the statistics of overall layers. This statistical observation indicates that MoE FFN layers would be well suited for the quantization.

Based on the observation, the FFN weights are following a normal distribution with a mean value near zero. Therefore, we use symmetric quantization without needing to shift the zero point. Even for the dense FFN layers, the means and the standard deviations are around zero except for the outliers which can be seen in the box plot of Figure 1. This symmetric quantization also gives an advantage to quantize many weight values near center to zero which could result in a sparse model weight.

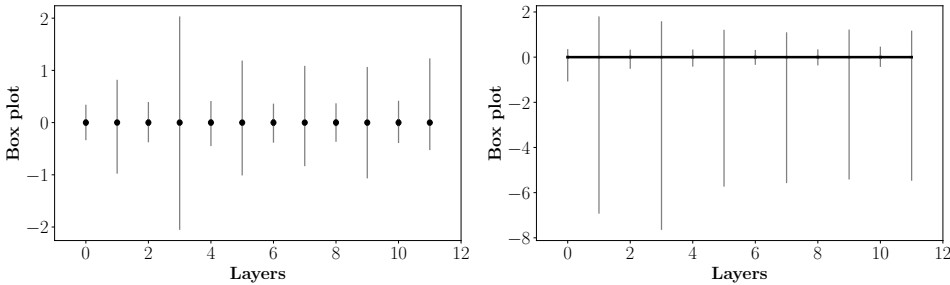

(a) FFN linear 1 weight distribution across layers    (b) FFN linear 2 weight distribution across layers

Figure 1: FFN weight distribution across layers. Even number layers $\{0, 2, ...\}$ are expert FFN layers and odd number layers $\{1, 3, ...\}$ are normal dense FFN layers. (a) shows the first linear layer in FFN and (b) shows the second linear layer in FFN.

### 3.2 QUANTIZATION ALGORITHMS

#### 3.2.1 QUANTIZATION TECHNIQUES

We try two quantization techniques, they are (i) linear quantization which is mapping quantized integer values and the original float value uniformly and (ii) log-based quantization from Aji & Heafield (2020) which maps integer and float ranges in a log scale. In both cases, we choose channel-wise quantization over matrix-wise quantization based on the experiment in Appendix A.

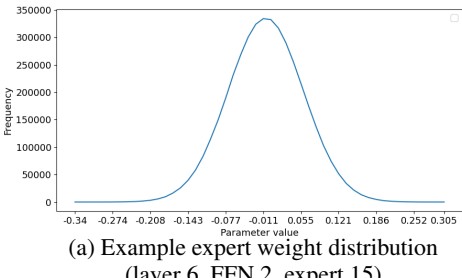 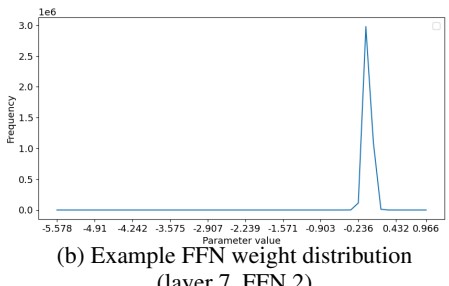

(a) Example expert weight distribution
(layer 6, FFN 2, expert 15)

(b) Example FFN weight distribution
(layer 7, FFN 2)

Figure 2: A comparison of example weight distributions from MoE and dense FFN layers.

**Linear quantization with absolute maximum.** The first technique is linear quantization which, given a matrix $\boldsymbol{A}$ and b bits, it encodes $\boldsymbol{A}$ as follows:

$$\boldsymbol{s}_j = \frac{2 \times \max(|\boldsymbol{A}_{:,j}|)}{2^b - 1}$$
$$\boldsymbol{Q}_{:,j} = \text{int}(\frac{\boldsymbol{A}_{:,j}}{\boldsymbol{s}_j})$$

where $s$ is the scaling factor which can be chosen per channel as shown or per the whole tensor. At inference time, the quantized $\boldsymbol{Q}$ is dequantized back to $\boldsymbol{A}'$ with the scaling factor $\boldsymbol{s}$ as follows:

$$\boldsymbol{A}'_{:,j} = \boldsymbol{Q}_{:,j} \times \boldsymbol{s}_j$$

**Log-scale quantization.** The second technique is log-scale quantization where 1 bit is kept for the sign and $(b-1)$ bits are used to encode the log-scaled values. Given a matrix $\boldsymbol{A}$, the quantization formula is as follows:

$$\boldsymbol{P} = sign(\boldsymbol{A})$$
$$\boldsymbol{T} = clip(\frac{|\boldsymbol{A}|}{s}, 1, 2^{1-2^{b-1}})$$
$$\boldsymbol{Q} = \lceil log_2(\frac{2}{3}\boldsymbol{T}) \rceil$$

where $s$ can be chosen in two ways, either (i) the absolute maximum or (ii) the optimal value to minimize the mean squared error (MSE) between the quantized and original values which is described in Aji & Heafield (2020). We use the second algorithm which we observe a better accuracy with the quantization. At inference time, the quantized weight values are dequantized based on the formula as follows:

$$\boldsymbol{A}' = \boldsymbol{P} \times s \times 2^{\boldsymbol{Q}}$$

**Comparison of quantization techniques.** Figure 3 shows the comparison between two quantization techniques with low bits applied on expert FFN layers and dense FFN layers. For dense FFN layers, log-scale quantization performs slightly better, but both do not work well on 2-bit resulting in almost zero evaluation scores. For expert FFN layers, both techniques work similarly for 3 and 4 bits, but log-scale quantization loses the accuracy seriously with 2-bit. This is because there are only 4 bins for the integer values to quantize with 2-bit quantization and one of them is zero. Log-scale tries to split values near zero in a more fine-grained way, but this actually hurts the performance compared to having enough zeros with linear quantization. Based on this experiment, we use linear quantization for compressing MoE FFN layers.

### 3.2.2 ROBUSTNESS OF EXPERT LAYERS TO QUANTIZATION

To better understand how applying quantization on different parts of an MoE model affects the accuracy, we conduct a set of experiments with various quantization bits. We divide an MoE model

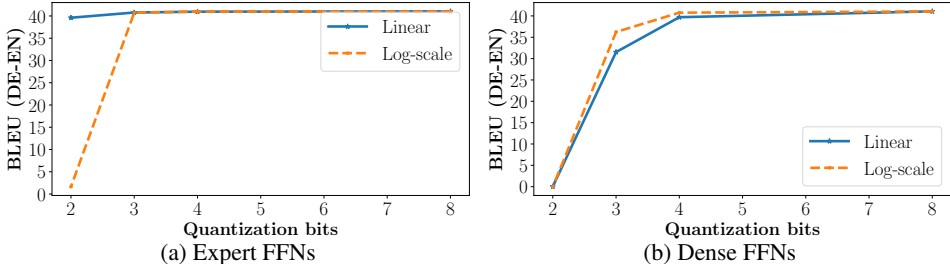

Figure 3: Linear quantization vs log-scale with optimal $s$ quantization.

into four parts: (i) expert FFNs, (ii) dense FFN layers, (iii) self-attention layers and (iv) cross-attention layers. Based on the observation that linear quantization works better with lower bits, we use it for this set of experiments.

Figure 4 shows evaluation BLEU scores when quantizing different parts of the MoE model. We observe that quantizing expert FFN layers to 2-bit does not seriously impact the overall model quality. However, quantizing other parts of the model into 2-bit hurts the output quality significantly. Quantized cross-attention and self-attention blocks still can maintain the quality with 3-bit quantization, but their performance gets impacted with 2-bit quantization. On the other hand, dense FFN layers get significant impact with lower bit quantization of 2-bit and 3-bit. With 3-bit quantization, the model score drops 23 % of original score, and 2-bit quantization on dense FFN layers gives almost zero score. We also include the same study on a dense model in Appendix B, the similar pattern with 2 and 3 bit quantization is observed.

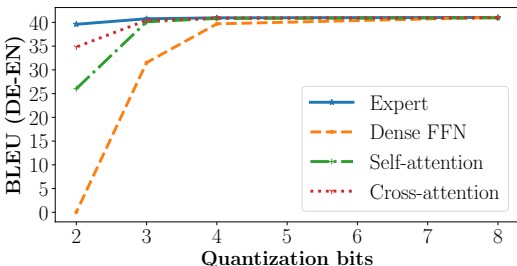

Figure 4: Quantization impact on different MoE model parts (channel-wise linear quantiztation without any additional training).

### 3.3 MIXTURE OF QUANTIZED EXPERTS (MOQE)

Based on the experiments from the previous parts of this section, we propose a very simple, highly effective and accurate quantization recipe for MoE models.

- Apply **weight-only quantization** while keeping activations in fp16.
- Quantize **expert FFN layers only**.
- Use **channel-wise and symmetric quantization**.
- Choose one of either two quantization methods depending on the quantization precision
    1. (3-bit or higher bit): Directly quatize trained MoE models without additional calibration.
    2. (2-bit): Fine-tune the model with Quantization Aware Training (QAT) which the descbtion follows.

**Sparse experts with quantization aware training** QAT is a well-known method used to recover the accuracy loss from the quantization (Gholami et al., 2022). In our case, to quantize to 2-bit precision, we can continue training the model with the original training data while applying quantization only on the forward pass computation as presented in (Wu et al., 2020; Bengio

et al., 2013) for recovering the accuracy loss. As we use symmetric quantization with 2-bit, zero numerical value is always included. Due to the normal distribution of expert weights centered around zero, many weight values naturally turn into zeros. This procedure results in sparse expert matrices.

# 4 EXPERIMENTS

## 4.1 EXPERIMENTAL SETUP

**Task.** We use multilingual machine translation task for our experiments with two different dataset which are 20 language directions and 10 language directions respectively. We also evaluate the proposed method on a different task presented in Appendix D. We use sacrebleu [2] on the detokenized output to measure the accuracy of the models. A single NVIDIA PCIE V100 running inside a docker container running Ubuntu 20.04 and CUDA 11.6 is used for all experiments, and all code is compiled with nvcc and gcc/g++ 9.3. We measure end-to-end runtime of the inference for the evaluation dataset.

**Datasets.** We use two different datasets described below. For the larger dataset setting, we use internally collected dataset consists of 6 different languages which are German (de), French (fr), Italian (it), Spanish (es), Dutch (nl) and English (en). They are crawled from web, and each language pair has at least several hundred million sentences. We use 128,000 sub-words vocabulary built with sentencepiece [3] library. The number of training sentences is included in Appendix G.
For the smaller dataset setting, we use WMT-10 benchmark dataset widely used for public benchmarks (Wang et al., 2020; Kim et al., 2021). There are 32.5 million sentence pairs for English-centric 20 language pairs including French (fr), Czech(cs), German (de), Finnish (fi), Latvian (lt), Estonian (et), Romanian (ro), Hindi (hi), Turkish(tr) and Gujarati (gu).

**Model architecture.** For all the experiments with large dataset, we use 24 transformer (Vaswani et al., 2017) encoder layers and 12 transformer decoder layers following the deeper encoder and shallower decoder practice (Kim et al., 2019; Kasai et al., 2021) to be more efficient at auto-regressive decoding. The embedding dimension is $1,024$ and FFN hidden dimension is $4,096$. For the positional information encoding to the hidden state, we use Transformer with Untied Positional Encoding (TUPE) proposed in Ke et al. (2021) instead of the conventional sinusoidal positional embedding. Another design choice is the location of layer normalization. For the training stability, we use pre-layer normalization proposed in Xiong et al. (2020) instead of the original post-layer normalization from (Vaswani et al., 2017). We train MoE and dense models for the comparison. The model architecture choices mentioned here are common for both models. The only difference between dense and MoE models is the number of experts. We use 32 experts for the MoE model trained with the larger web data. We use beam search decoding with beam size of 5. For the experiments with smaller dataset, we use 12 transformer encoder layers and 6 transformer decoder layers. The embedding dimension is 768 and FFN hidden dimension is $3,072$. In this setting, we use MoE layers with 128 experts at every other layer.

**MoE architecture.** For the MoE model specific settings, we use top-1 learned gating from Fedus et al. (2021) and use an MoE layer at every other layer which are even numbered layers (Lepikhin et al., 2020; Fedus et al., 2021; Kim et al., 2021). During the training of MoE models, we use jittering noise and balancing loss (ratio of $0.01$) suggested in Lepikhin et al. (2020); Fedus et al. (2021) to more uniformly distribute expert utilization. To prevent overfitting and better regularize the model, we use gating dropout ($0.2$) (Liu et al., 2022) as well.

## 4.2 MOQE PERFORMANCE RESULTS

We apply MoQE quantization recipe to an MoE model and compare the performance with several dense models in Table 2. This experiment is done on the larger web dataset. The baseline is a dense model trained on the same dataset as the MoE model. Throughput, memory size and sparsity are all measured with the fp16 precision model. As additional comparison points, the dense model is also

---

[2]https://github.com/mjpost/sacrebleu
[3]https://github.com/google/sentencepiece

Table 2: The model performance comparison. All the models are trained on same data up to the convergence with 200,000 update steps. The baseline is the FLOPs equivalent dense model's BLEU score and speed.

| Model type | Precision | Average BLEU (difference %) | Throughput (X times) | Size (X times) | Sparsity % |
|---|---|---|---|---|---|
| Dense (baseline) | fp16 | 45.06 (0) | 1X | 1X | 3.8e-5 |
| Dense | int8 | 45.04 (-0.02) | Not optimized | 0.88X | 1.28 |
| | int4 | 44.84 (-0.47) | Not optimized | 0.82X | 21.31 |
| MoE 5.3B (32 experts) | fp16 | **46.35 (+2.87)** | 0.38X | 8.38X | 3.8e-5 |
| | int8 | **46.34 (+2.85)** | 1.00X | 4.57X | 1.24 |
| | int4 | 46.18 (+2.49) | **1.03X** | 2.67X | 20.68 |
| | int3 | 46.01 (+2.11) | Not optimized | 2.19X | 42.15 |
| | int2 QAT | 45.90 (+1.88) | Not optimized | 1.71X | 79.10 |

quantized to 8-bit and 4-bit only on the even numbered FFN layers which is the best configuration for quantizing the dense model described in Appendix B. For the MoE model, various quantization settings ranging from 8-bit to 2-bit are measured together with the original fp16 performance. For 2-bit quantization, additional QAT is applied. Finally, we applied magnitude based pruning approach to the 2-bit quantized model to acquire a sparser model.

First of all, the MoE model achieves 2.87% improvement on the BLEU score while increasing the model size to 8.38X of the original dense model. When 4-bit post-training quantization is applied, it still maintains 2.11% higher BLEU score than the original dense model. And, it could achieve even faster speed than the dense model which is 2.7X speed-up from the fp16 MoE model. This also reduces the memory consumption significantly from 8.38X to 2.67X compared to the dense model. With 2-bit QAT, the MoE model can still maintain 1.88% higher quality than the original dense model, but the model size is now only 1.71X of the original dense model. Also, the matrices are sparse up to 79.1% of the values are zeros.

Figure 5 shows the sparsity distribution of different layers. The second linear layers after the non-linear activation layers show higher sparsity compared to the first linear layers. Some layers could reach up to 85% sparsity. We include a further investigation of sparsity with magnitude based pruning approach in Appendix I.

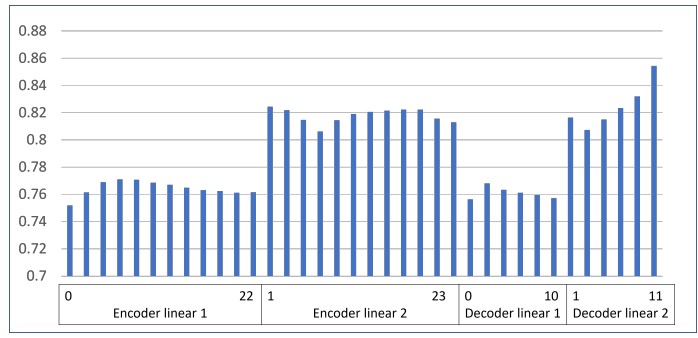

Figure 5: Sparsity distribution of 2-bit quantized MoE layers.

## 4.3 ROBUSTNESS COMPARISON BETWEEN MOE AND DENSE MODELS

We compare robustness against low-bit quantization between MoE and dense models using the post-training quantization without any QAT. For the dense model, quantization with different bits is applied to the even numbered FFN layers. Appendix B shows this is the best layer selection for the dense model. We use two different datasets to verify the proposed quantization method works in different model settings.

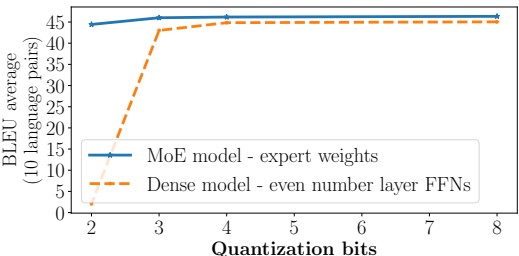

Figure 6: Quantization performance comparison between MoE and dense models. 10 different language pair scores are averaged.

Figure 6 presents the experiment with the model trained with the larger dataset. It shows the average BLEU scores with different quantization precision for both MoE and dense models. The MoE model can maintain accuracy within -0.3 down to 3-bit and -1.82 for 2-bit. On the other hand, the dense model can preserve the accuracy only down to 4-bit, but starts to lose significant accuracy more than 2 BLEU scores when it goes down to 3-bits. In case of 2-bits, dense model loses most of capability by -42.96 BLEU scores. Table 9 shows the score differences by quantization for both MoE and dense models on 10 different language pairs translations.

Figure 7 presents the experiment with the model trained with the smaller dataset. In this setting, each individual expert is smaller, but there are 4 times more experts in one MoE layer. And, they are trained with smaller dataset, so they do not have equivalent knowledge as the previous model trained on the larger dataset. As can be seen in the Figure, the quantization performance shows a similar pattern. The MoE model preserves accuracy even when it is quantized to 2 or 3 bits. However, dense model quickly loses the performance when it is quantized down to lower than 4-bit. Again, the MoE model is much more robust to quantization than the dense model.

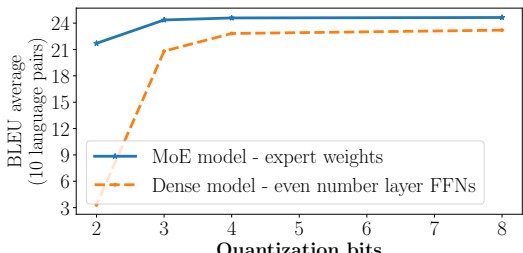

Figure 7: Quantization performance comparison between MoE and dense models. 20 different WMT language pairs are averaged.

## 5 CONCLUSION AND FUTURE WORKS

This paper shows how much MoE models are robust to the low-bit quantization with various experiments. By analyzing component-wise sensitivity and various quantization design choices, we present an efficient and effective way to reduce the model size which results in 4.9X model size reduction. With an optimized runtime, 4-bit quantized model can run 2.71X faster than the fp16 model. We also show 2-bit quantization could achieve more than 79% sparsity in the expert weights. The results naturally bring interesting future research directions. The discovered robustness of expert layers can guide a better way to train MoE models. If we can better control the splitting of latent space, better MoE models can be acquired. Analyzing the interactions between expert FFN layers and the other common layers in the model could guide a way to build a composable model. Especially, as presented in Appendix E, we observe that quantization sometimes improves the accuracy on tasks in a specific situation. Another important direction will be studying how to accelerate sparse expert computation on modern hardware with software/hardware co-design. This will eventually make MoE models much more efficient in both training and inference.

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

## A    CHANNEL-WISE VS MATRIX-WISE QUANTIZATION

Scaling factors are calculated by the quantization algorithm and stored in half precision floating-point (fp16) numbers to dequantize the matrices with. These factors can be chosen on the channel scale or the whole matrix scale. As shown in figure 8, channel-wise quantization gives quite higher scores than tensor-wise especially for low precision. Additional parameters to store channel-wise scaling factors is small, because only one value is needed for a channel and less than 1% of total parameters in a matrix. Therefore, we use channel-wise quantization for all the quantization experiments.

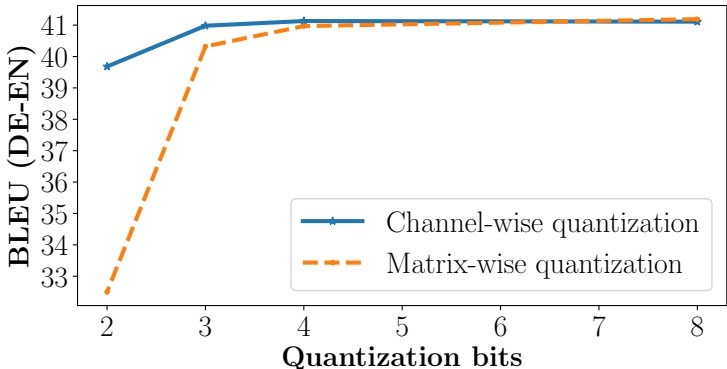

Figure 8: Linear quantization of expert FFNs with channel-wise and matrix-wise scaling factors.

## B    QUANTIZATION OF DIFFERENT LAYERS IN A DENSE MODEL

In the paper, we compare a dense model and an MoE model in terms of quantization robustness. To make a fair comparison, we consider quantizing only half of the dense transformer blocks' FFNs, because we quantize expert weights only which exist only in every other block (even numbered). We compare three different configurations - (1) quantizing even numbered blocks' FFNs only, (2) quantizing odd numbered blocks' FFNs only and (3) quantizing all FFN layers. As can be seen in Figure B, quantizing even numbered blocks' FFNs affects the accuracy the least, and quantizing all FFN layers give the worst result. Based on this experiment, we quantize only even numbered transformer blocks' FFNs for the dense model in all the experiments and comparisons.

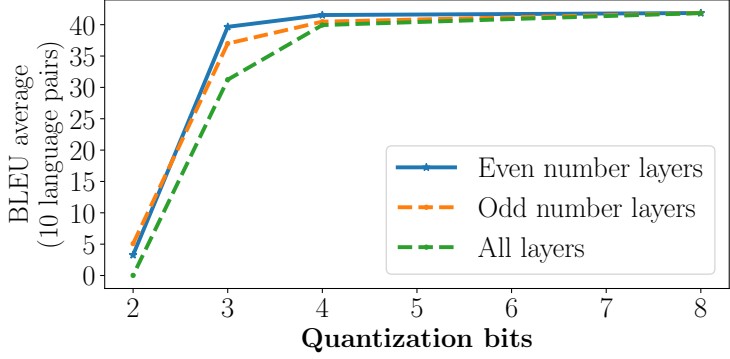

Figure 9: Quantization impact of different layers in a dense model.

## C    SKEWNESS OF WEIGHT MATRICES IN MOE AND DENSE MODELS

In the analysis of model weight distribution in Section 3, we observe that dense models' FFN layers tend to have more outliers than MoEs' expert FFN layers. We measure the skewness of weight distribution of those in Table 3.

Table 3: Expert vs non-expert FFN layers parameters distribution skewness

| Parameter | skew |
|---|---|
| encoder expert 15 FFN fc1 layer 0 | -0.002 |
| encoder expert 15 FFN fc2 layer 0 | -0.190 |
| encoder expert 15 FFN fc1 layer 6 | -0.002 |
| encoder expert 15 FFN fc2 layer 6 | -0.002 |
| encoder non-expert FFN fc1 layer 1 | -0.019 |
| encoder non-expert FFN fc2 layer 1 | -10.729 |
| encoder non-expert FFN fc1 layer 7 | 0.003 |
| encoder non-expert FFN fc2 layer 7 | -1.574 |
| encoder expert FFN fc1 mean | 0.00 |
| encoder expert FFN fc2 mean | -0.63 |
| decoder expert FFN fc1 mean | 0.00 |
| decoder expert FFN fc2 mean | 0.48 |
| encoder non-expert FFN fc1 mean | 0.00 |
| encoder non-expert FFN fc2 mean | -1.84 |
| decoder non-expert FFN fc1 mean | 0.00 |
| decoder non-expert FFN fc2 mean | -0.09 |

## D   ABSTRACTIVE SUMMARIZATION TASK PERFORMANCE

To validate the quantization performs well on a different task and a model, we evaluate a 10.1 B MoE (64 experts) model's quantization performance on an abstractive summarization task called **XSUM** (Narayan et al., 2018). Table 4 shows that the MoE model performs well with low-bit quantizations such as 2-bits and 3-bits.

Table 4: The ROUGE score differences in percentage (%) after applying post-training quantization on XSUM evaluation.

| Quantization Bits | R1 | R2 | RL |
|---|---|---|---|
| 8-bit | +0.17 | +0.13 | +0.26 |
| 4-bit | -0.10 | -0.26 | -0.34 |
| 3-bit | -0.21 | -0.82 | -0.28 |
| 2-bit | -1.61 | -5.94 | -2.37 |

## E   BETTER GENERALIZATION WITH EXPERT QUANTIZATION

We observe an interesting phenomenon that quantization actually improves the score of evaluation on a different domain dataset. We trained an MoE model with 64 experts (10.1B) on 50 different language translations (98 English-centric language pairs). When we evaluate this model on a different domain subset 6 languages (German, Spanish, French, Italian, Dutch, English), the evaluation BLEU score increases until we quantize the experts down to 3-bits without any additional QAT or calibrations. With 3-bit quantization, the score increases more than 6.42% on non-English to English and 6.96% on English to the others. Especially, from English to Italian and from Italian to English scores increase more than 10% which quite significant. The results are summarized in Table 5. We are analyzing what could be the reason for this phenomenon, but we think this is related to how MoE models learn representations. MoE layers might learn very specific knowledge with its increased capacity, but the shared layers learn more generic knowledge. By blurring the representation from the MoE layers, the model becomes more general task capable. This is one of our future research areas.

Table 5: The BLEU score differences in percentage (%) after quantization on different language pairs.

| Quantization Bits | de-en | es-en | fr-en | it-en | nl-en | Average (xx-English) |
|---|---|---|---|---|---|---|
| fp16 | 40.43 | 47.83 | 46.46 | 39.67 | 43.13 | 43.50 |
| 8-bit | **-0.01** | -0.08 | +0.17 | +0.25 | +0.06 | +0.08 |
| 4-bit | -0.04 | +5.60 | +2.83 | +7.72 | +5.39 | +4.30 |
| 3-bit | -0.47 | **+9.22** | **+4.02** | **+11.97** | **+7.38** | **+6.42** |
| 2-bit | -7.59 | -0.30 | -7.52 | -4.45 | -4.34 | -4.84 |
| | **en-de** | **en-es** | **en-fr** | **en-it** | **en-nl** | **Average (English-xx)** |
| fp16 | 36.74 | 40.98 | 45.99 | 29.13 | 38.72 | 38.31 |
| 8-bit | -0.03 | +0.13 | -0.19 | +0.06 | +0.29 | +0.05 |
| 4-bit | **+0.88** | +6.86 | +3.55 | +6.52 | +2.59 | +4.08 |
| **3-bit** | +0.25 | **+9.65** | **+5.24** | **+16.09** | **+3.57** | **+6.96** |
| 2-bit | -19.58 | -9.11 | -6.99 | -2.22 | -14.68 | -10.52 |

## F  ADDITIONAL SPARSITY DISTRIBUTION

We include sparsity distribution across different layers in a dense model quantized with 4-bit. As can be seen in Figure 10, the sparsity is overall low. Second linear layers in FFN show slightly higher sparsity, but all of them are smaller than 30%.

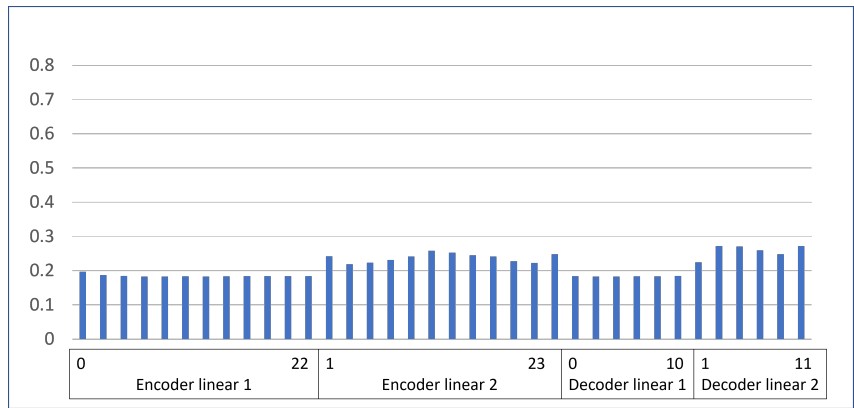

Figure 10: Sparsity distribution of 4-bit quantized dense layers.

## G  MACHINE TRANSLATION DATASET SUMMARY

Table 6 shows the number of parallel sentences used to train dense and MoE models. All languages have at least 300 million sentences and the differences in the number among languages are less than two times.

Table 6: The number of parallel sentences including backtranslation data.

| Language | Number of parallel sentences (million) | |
|---|---|---|
| | xx → English | English → xx |
| DE (German) | 505 | 411 |
| ES (Spanish) | 448 | 407 |
| FR (French) | 448 | 376 |
| IT (Italian) | 447 | 303 |
| NL (Dutch) | 302 | 378 |

## H    QUANTIZATION AWARE TRAINING (QAT)

For the QAT with straight through estimator, we use the hyper-parameters as in Table 7. Figure 11 shows the validation loss curve of one training run with 2-bit expert quantization.

Table 7: Expert vs non-expert FFN layers parameters distribution skewness

| Parameter Name | Value |
|---|---|
| Start learning rate | 0.001 |
| Warm-up steps | 1000 |
| Effective batch size | 1.5M tokens |
| Optimizer | Adam |
| Betas (Adam) | (0.9, 0.999) |
| Weight decay (Adam) | 0 |
| Gradient clipping | 0.1 |
| Training steps | 60,000 |

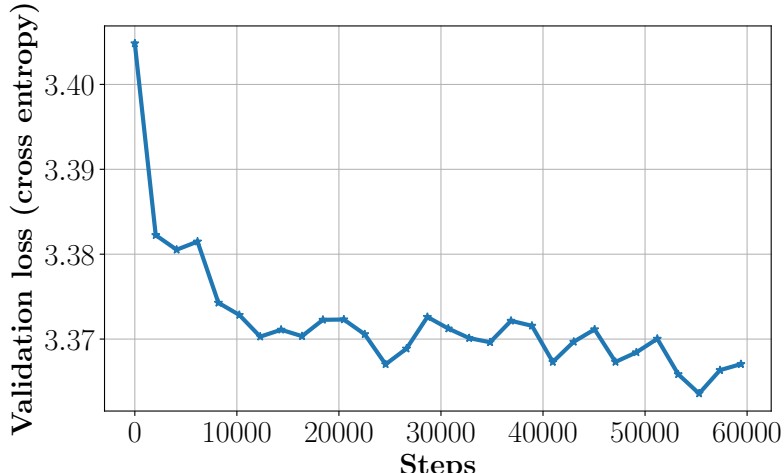

Figure 11: Validation loss curve for QAT.

## I    MAGNITUDE PRUNING EXPERIMENTS

Inspired by the emerged sparsity in expert layers, we apply a simple magnitude based pruning to the MoE model we experiment with. We apply different threshold values from 0.00001 to 0.5. We make all the weight values less than the threshold to be zero. We apply 2 to 8 bit quantization together. Figure 12 shows how model performance varies with the achieved sparsity. Even with sparsity level of 90%, the model preserves a certain level of task capability. Compared to Gale et al. (2019), this shows much higher performance with the same sparsity. This could be another example showing the robustness of expert weights.

## J    DETAILED BLEU SCORE DIFFERENCES WITH QUANTIZATION APPLIED TO THE MODEL TRAINED ON PUBLIC WMT DATASET

Table 8 shows individual BLEU score changes with various quantization bits for MoE and dense models trained on public WMT dataset.

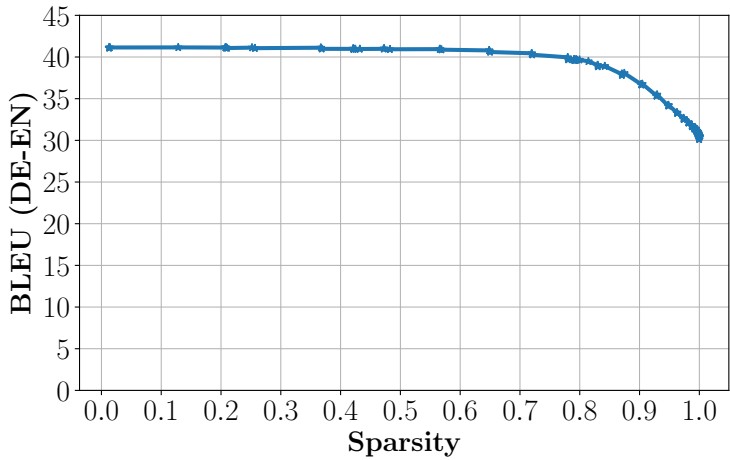

Figure 12: Sparsity of MoE layers vs. BLEU scores with magnitude pruning.

Table 8: The BLEU score differences in percentage (%) after quantization on different language pairs in WMT dataset. The rows with fp16 show the baseline BLEU scores.

| Bits | Model | en-cs | en-de | en-et | en-fi | en-fr | en-gu | en-hi | en-lv | en-ro | en-tr | Avg.(en-xx) |
|---|---|---|---|---|---|---|---|---|---|---|---|---|
| fp16 | Dense | 23.89 | 31.46 | 17.80 | 18.75 | 28.54 | 10.34 | 11.98 | 22.29 | 27.22 | 15.81 | 20.81 |
| (BLEU) | MoE | 26.09 | 34.36 | 18.27 | 22.17 | 31.34 | 13.04 | 12.16 | 23.26 | 27.95 | 16.89 | 22.55 |
| 8-bit | Dense | -0.39 | -0.09 | -0.32 | 0.60 | 0.01 | -0.80 | 0.61 | -0.26 | 0.17 | -0.09 | -0.05 |
| | MoE | 0.01 | -0.15 | 0.64 | -0.33 | 0.19 | 0.86 | 0.02 | -0.04 | -0.15 | -0.03 | 0.05 |
| 4-bit | Dense | -1.11 | -1.91 | -3.15 | -1.50 | 1.03 | -7.08 | -4.44 | -2.38 | -1.65 | -1.89 | -1.90 |
| | MoE | -0.30 | -0.62 | 0.30 | -0.62 | -0.13 | -0.97 | 1.53 | -0.81 | -0.82 | -0.22 | -0.36 |
| 3-bit | Dense | -10.87 | -7.86 | -12.87 | -11.70 | -3.96 | -32.03 | -24.76 | -11.16 | -7.05 | -12.74 | -11.24 |
| | MoE | -0.84 | -1.06 | -1.79 | -1.97 | 0.35 | -2.80 | -0.70 | -1.98 | -1.05 | -1.64 | -1.21 |
| 2-bit | Dense | -97.44 | -86.29 | -91.79 | -91.02 | -85.75 | -98.26 | -96.48 | -94.14 | -87.30 | -95.02 | -91.21 |
| | MoE | -8.84 | -9.15 | -17.06 | -13.24 | -5.62 | -25.24 | -16.38 | -16.11 | -11.04 | -14.48 | -12.34 |
| Bits | Model | cs-en | de-en | et-en | fi-en | fr-en | gu-en | hi-en | lv-en | ro-en | tr-en | Avg.(xx-en) |
| fp16 | Dense | 29.48 | 35.62 | 23.43 | 23.91 | 31.89 | 16.54 | 14.97 | 26.25 | 35.68 | 18.52 | 25.63 |
| (BLEU) | MoE | 31.25 | 38.21 | 23.67 | 25.64 | 32.59 | 19.55 | 15.89 | 25.22 | 34.80 | 20.27 | 26.71 |
| 8-bit | Dense | 0.02 | -0.02 | 0.10 | -0.33 | -0.15 | -0.37 | -0.40 | 0.33 | -0.34 | 0.14 | -0.09 |
| | MoE | 0.07 | 0.12 | 0.08 | 0.06 | -0.10 | 0.14 | -0.49 | -0.03 | 0.05 | -0.17 | 0.00 |
| 4-bit | Dense | -0.24 | -0.78 | -3.74 | -1.72 | -1.69 | -4.58 | -0.56 | -1.97 | -0.15 | -1.84 | -1.53 |
| | MoE | 0.44 | 0.01 | -1.00 | 0.25 | -0.03 | 0.07 | 1.06 | -0.98 | 0.67 | -0.56 | 0.01 |
| 3-bit | Dense | -7.25 | -7.11 | -10.44 | -10.36 | -6.44 | -18.67 | -16.68 | -11.52 | -7.39 | -10.39 | -9.68 |
| | MoE | -0.86 | -0.14 | -2.04 | -1.10 | 1.02 | -2.55 | 1.11 | -2.11 | -1.45 | -2.91 | -1.01 |
| 2-bit | Dense | -81.78 | -74.17 | -83.08 | -85.13 | -72.44 | -94.23 | -89.54 | -81.50 | -80.54 | -85.70 | -81.33 |
| | MoE | -6.12 | -7.69 | -16.78 | -11.29 | -2.16 | -20.14 | -16.42 | -15.82 | -12.34 | -17.61 | -11.54 |

## K DETAILED BLEU SCORE DIFFERENCES WITH QUANTIZATION APPLIED TO 5.3B MODEL.

Table 9 shows individual BLEU score changes with various quantization bits for MoE and dense models measured with the internal validation dataset. Table 10 shows the same model's evaluation performance on two WMT public dataset.

Table 9: The BLEU score differences in percentage (%) after quantization on different language pairs. The rows with fp16 show the baseline BLEU scores.

| Quantization Bits | Model | de-en | es-en | fr-en | it-en | nl-en | Avg. (xx-English) |
|---|---|---|---|---|---|---|---|
| fp16 | Dense | 40.31 | 53.09 | 49.13 | 44.03 | 46.23 | 46.56 |
| (Baseline BLEU) | MoE | 41.49 | 53.79 | 50.26 | 46.97 | 47.53 | 48.01 |
| 8-bit | Dense | -0.03 | -0.08 | -0.02 | 0.01 | -0.05 | -0.04 |
| (% difference) | MoE | -0.10 | -0.06 | 0.00 | -0.02 | 0.03 | -0.03 |
| 4-bit | Dense | -0.78 | 0.29 | -0.23 | -0.93 | -0.20 | -0.37 |
| (% difference) | MoE | -0.50 | -0.11 | -0.10 | -0.39 | -0.02 | -0.22 |
| 3-bit | Dense | -6.36 | -2.51 | -4.24 | -5.93 | -2.67 | -4.34 |
| (% difference) | MoE | -0.92 | 0.26 | -0.26 | -1.26 | 0.29 | -0.38 |
| 2-bit | Dense | -95.44 | -94.42 | -95.51 | -95.10 | -93.31 | -94.76 |
| (% difference) | MoE | -4.35 | -1.00 | -2.64 | -7.01 | -0.70 | -3.14 |
| | | en-de | en-es | en-fr | en-it | en-nl | Avg. (English-xx) |
| fp16 | Dense | 38.74 | 46.44 | 50.82 | 40.09 | 41.69 | 43.55 |
| (Baseline BLEU) | MoE | 39.90 | 47.47 | 52.45 | 41.25 | 42.36 | 44.69 |
| 8-bit | Dense | -0.04 | -0.07 | 0.02 | -0.05 | 0.09 | -0.01 |
| (% difference) | MoE | 0.05 | -0.01 | -0.03 | 0.00 | 0.00 | 0.00 |
| 4-bit | Dense | -0.76 | -1.11 | -0.29 | -0.70 | -0.26 | -0.62 |
| (% difference) | MoE | 0.31 | -0.90 | -0.74 | -0.45 | -0.68 | -0.49 |
| 3-bit | Dense | -5.82 | -4.79 | -3.96 | -5.41 | -4.54 | -4.91 |
| (% difference) | MoE | -0.21 | -2.12 | -1.41 | -0.87 | -0.89 | -1.10 |
| 2-bit | Dense | -97.28 | -96.16 | -95.52 | -96.68 | -94.83 | -96.09 |
| (% difference) | MoE | -5.24 | -6.19 | -5.19 | -5.30 | -4.48 | -5.28 |

Table 10: The BLEU score differences in percentage (%) of 5.3B MoE model after quantization on different language pairs on WMT datasets. The rows with fp16 show the baseline BLEU scores.

| Quantization Bits | Model | de-en | fr-en | Avg. (xx-English) |
|---|---|---|---|---|
| fp16 | Dense | 50.11 | 42.98 | 46.54 |
| (Baseline BLEU) | MoE | 52.73 | 44.04 | 48.39 |
| 8-bit | Dense | 0.04 | 0.11 | 0.07 |
| (% difference) | MoE | 0.09 | -0.04 | 0.03 |
| 4-bit | Dense | -0.59 | -1.27 | -0.91 |
| (% difference) | MoE | -0.47 | -0.36 | -0.42 |
| 3-bit | Dense | -5.75 | -6.17 | -5.94 |
| (% difference) | MoE | -1.15 | -0.90 | -1.03 |
| 2-bit | Dense | -96.88 | -95.59 | -96.28 |
| (% difference) | MoE | -5.37 | -3.68 | -4.60 |
| | | en-de | en-fr | Avg. (English-xx) |
| fp16 | Dense | 50.90 | 44.47 | 47.68 |
| (Baseline BLEU) | MoE | 52.90 | 45.51 | 49.21 |
| 8-bit | Dense | 0.00 | 0.02 | 0.01 |
| (% difference) | MoE | -0.05 | 0.23 | 0.08 |
| 4-bit | Dense | 0.24 | -1.31 | -0.48 |
| (% difference) | MoE | -0.93 | 0.25 | -0.39 |
| 3-bit | Dense | -5.86 | -7.53 | -6.64 |
| (% difference) | MoE | -1.41 | -0.69 | -1.08 |
| 2-bit | Dense | -97.77 | -96.22 | -97.05 |
| (% difference) | MoE | -6.34 | -6.15 | -6.25 |

