# OpenReview forum: "Mixture of Quantized Experts (MoQE): Complementary Effect of Low-bit Quantization and Robustness"
_ICLR.cc/2023/Conference — Submitted to ICLR 2023_

### Official Review · Reviewer_P5VH · 2022-10-24

**Confidence:** 4
**Correctness:** 3
**Technical Novelty And Significance:** 2
**Empirical Novelty And Significance:** 3
**Recommendation:** 6

**Clarity, Quality, Novelty And Reproducibility:**

Novelty is limited as existing techniques and architectures are leveraged. However, the set of experiment performed and the resulting observations are new and interesting.

Readability is fine but the manuscript would benefit from extensive polishing and proofreading. Some recommendations:
- section 2: fix "the model requires much more memory to load the model"
- section 2: fix "the actual inference speed decreases slower than 40%"
- section 2 lists 4 challenges to MoE deployment. It seems to me the challenges are two (or three): memory footprint and slower training/inference. The latter is caused by communication overhead and bandwidth bottleneck. I recommend reorganizing this section
- paper structure could also be improved:
  - section 3 describes quantization of specific layers and discusses results before introducing the model architectures (Section 4)
  - table 1 shows MoE weight % for an unspecified "specific model setting", which is later introduced in Section 4
- section 3: weight distributions in fig 2a,b could be presented on a semi-log scale to highlight the outliers (maybe combining the two figures into one)
- section 3.3 mentioned QAT for the first time and introduces it in the process flow for 2 bits quantization based on "the experiments from the previous parts of this section". Are results shown in Fig. 4 obtained with PTQ? This is never mentioned.
- multiple figure labels read "qunatization bits"

**Strength And Weaknesses:**

Strengths:
- paper convincingly demonstrates that aggressive quantization of the expert FNN retains significant accuracy improvement while limiting the increase in model size
- ablation studies give valuable information on where quantization is best applied (expert FNN)
- observation of high sparsity resulting from aggressive quantization may also be leveraged by future studies

Weaknesses:
- primarily an observational study which applies known quantization techniques to known architectures
- a trade-off remains between model size and accuracy improvement (i.e., the MoE model is still larger than corresponding dense models, especially when compared to the quantized versions)
- only weights are quantized, not activations. Consequently, inference speedup is necessarily limited
- several throughput values on Table 2 are missing for both the dense and MoE model. What's the reason?
- as a result of the authors attempting to emphasize their best results, some statements can be misleading. For example, the conclusions read "We also show 2-bit quantization could achieve _more_ than 80% sparsity in the expert weights" (emphasis mine) which refers to the 2-bit pruned results which comes at the expense of losing most MoE gains in accuracy. I would recommend to tune down this and similar statements


**Summary Of The Paper:**

This paper presents an investigation of Mixture of Experts (MoE) transformer models with aggressive quantization (2-4 bits) of the Feed Forward Networks weights. 2-bit quantization of these layers limits model size increase due to MoE usage to 1.7x of the original model (instead of 8.4x), while retaining some of the higher-precision MoE accuracy gains (+1.88% instead of +2.87%). 2-bit quantization also results in significant sparsity of the FNN layers (~80%).

**Summary Of The Review:**

This paper presents an empirical investigation that applies known quantization techniques to transformers models with MoE. Although from this perspective novelty is limited, results are interesting as the authors identify a path towards aggressive quantization of these models and their observations may be leveraged in future works.

---

> ### Author Response · Authors · 2022-11-17
> **Response to Reviewer P5VH**
>
> Thank you so much for the detailed and very valuable comments.
>
> * *“observation of high sparsity resulting from aggressive quantization may also be leveraged by future studies”* \
> *“ablation studies give valuable information on where quantization is best applied (expert FNN)”* \
> **Indeed, we are excited about the findings which could guide us to design a better MoE architecture and training methods by exploiting these special characteristics.**
>
> * *"primarily an observational study which applies known quantization techniques to known architectures"* \
> **As mentioned in the general response, we focus more on practicality. We will clarify this in the writing more.**
>
> * *"a trade-off remains between model size and accuracy improvement (i.e., the MoE model is still larger than corresponding dense models, especially when compared to the quantized versions)"* \
> **Yes, we agree that there remains a trade-off. However, +1.88% BLEU score gain on 10 language directions is significant considering less than 2 times of model size difference. Up to int4 quantization, we could maintain +2.49% of improvement.**
>
> * *"only weights are quantized, not activations. Consequently, inference speedup is necessarily limited"* \
> **We keep improving the algorithms and GPU kernels for low-bit quantized MoE operation. We will provide updated numbers in the revised paper which will give a better improvement.**
>
> * *"several throughput values on Table 2 are missing for both the dense and MoE model. What's the reason?"* \
> **First of all, due to the low quality compared to the MoE models, we do not optimize dense models. Without proper optimizations, the speed might be very slow, so we could not provide a fair comparison. For MoE models, we developed efficient kernels up to 4-bits so far. We are working on lower bits such as 3 and 2 bits. However, we wanted to focus on the behavior of the models in this paper.**
>
> * *"as a result of the authors attempting to emphasize their best results, some statements can be misleading. For example, the conclusions read "We also show 2-bit quantization could achieve more than 80% sparsity in the expert weights" (emphasis mine) which refers to the 2-bit pruned results which comes at the expense of losing most MoE gains in accuracy. I would recommend to tune down this and similar statements"* \
> **Thank you for the great suggestion. We will clarify this point in the revision. Actually, without pruning, we could observe 79.1% sparsity and still maintain quite good quality gain (+1.88%). With additional pruning, we could see greater than 80% sparsity, but this could be less meaningful than the naturally emerged sparsity.**
>
> * Proofreading recommendations \
> **Thank you for the detailed comments. We really appreciate them. We will incorporate those recommendations in the paper revision.**

---

### Official Review · Reviewer_ZfnQ · 2022-10-29

**Confidence:** 4
**Correctness:** 3
**Technical Novelty And Significance:** 2
**Empirical Novelty And Significance:** 2
**Recommendation:** 3

**Clarity, Quality, Novelty And Reproducibility:**

This paper mostly discussed the phenomena the authors observed after applying basic quantization techniques to MoE models. It would be more desirable to provide deeper insights on what can be further improved to innovate quantization performance for MoE models.

**Strength And Weaknesses:**

(Strength)
- One of the first efforts to investigate the quantization impact on MoE models


(Weakness)
- Although the observations are interesting, there is little technical innovation based on them. The proposed quantization schemes follow conventional quantization techniques.

- The authors claimed the gain in throughput, but the experimental settings for measuring hardware speedup are not clear.

- The evaluation of accuracy is quite limited (presented only one translation task).

**Summary Of The Paper:**

This paper investigated the characteristics of Mixture of Experts (MoE) models when their expert layers are quantized. The authors revealed that the expert layers have more evenly distributed data than dense layers, and thus they are more robust to quantization. Motivated by this observation, the authors applied low-bit quantization to MoE's expert layers and achieved memory savings with slight accuracy degradation.

**Summary Of The Review:**

The authors presented the discovery that the distributions of the expert layers of MoE models are less skewed than typical FFN layers, and thus the authors claimed that these expert layers are more friendly to quantization. Although these observations are interesting, it would be nicer to see further innovations in the quantization technique to squeeze more bits out of the MoE models.

---

> ### Author Response · Authors · 2022-11-17
> **Response to Reviewer ZfnQ**
>
> Thank you so much for the constructive comments.
>
> * *"Although the observations are interesting, there is little technical innovation based on them. The proposed quantization schemes follow conventional quantization techniques."* \
> **As we mention in the general response (1 & 2), we are focusing on the MoE layers' unique property and extensive analysis on it. Based on that, we propose a simple yet effective way to solve one of the hardest problems to practically deploy MoE models in real-world application. Because the simple quantization could preserve the most accuracy, we believe that we may not need to employ more complicated methods. Even though it's simple, we believe this would benefit many researchers in the field. Furthermore, the observations of emerging sparsity and robustness to the perturbation could guide us to a better design of MoE models.**
>
> * *"The authors claimed the gain in throughput, but the experimental settings for measuring hardware speedup are not clear."* \
> **Thanks for pointing this out. We will clarify this in the revised paper.**
>
> * *"The evaluation of accuracy is quite limited (presented only one translation task)."* \
> **As we mention in the general response 3, we actually conducted extensive experiments including a different task, different model sizes, and data. Due to the page limitation, we couldn't include them in the main body of the paper. We will clarify this fact in the paper.**

---

### Official Review · Reviewer_X5nA · 2022-10-30

**Confidence:** 4
**Correctness:** 2
**Technical Novelty And Significance:** 2
**Empirical Novelty And Significance:** 2
**Recommendation:** 3

**Clarity, Quality, Novelty And Reproducibility:**

- The presentation is not clear: The term MoQE is present in the title, but I cannot find another "MoQE" until Section 4.2. It is not clear to me what exactly MoQE means until Section 3.3.
- Section 3.3 talks about the recipe. However, a **significant portion** of details is missing. What is QAT? What is the post-training quantization? There is neither explanation nor reference on these terms.
- Novelty is incremental: the authors only combine existing quantization techniques with MoE layers. There is no new technique proposed except a quantization recipe.
- Some statements are not rigorous or lack explanation: for example, using jittering noise and balancing loss cannot uniformly distribute expert utilization (or at least need to show the patterns); it is not clear why symmetric quantization can give an advantage to quantize many weights near zero, since as long as the ''0'' bin covers many weights.

**Strength And Weaknesses:**

+ [+] The idea is straightforward enough
+ [+] The topic is timely
- [-] The presentation is not clear
- [-] The novelty is limited

Please refer to the next section for more details.

**Summary Of The Paper:**

This paper presents a quantization technique for MoE models. The authors first study the weight distributions of different layers (MoE layer, Dense layer) and found that the MoE layer has fewer weight outliers, which is why the authors claim that the MoE layers are more suited for quantization. The authors then studied two different quantization methods, i.e. linear quantization and log-scale quantization. The authors then show performance after quantization on different modules, and then propose a quantization recipe.

**Summary Of The Review:**

Given the limited novelty and unclear presentation, I currently give a score of 3. However, I am also open to change the score during the rebuttal time.

---

> ### Author Response · Authors · 2022-11-17
> **Response to Reviewer X5nA**
>
> Thank you so much for the constructive comments.
>
> * *"The presentation is not clear"* \
> *"The presentation is not clear: The term MoQE is present in the title, but I cannot find another "MoQE" until Section 4.2. It is not clear to me what exactly MoQE means until Section 3.3."*
> *"Section 3.3 talks about the recipe. However, a significant portion of details is missing. What is QAT? What is the post-training quantization? There is neither explanation nor reference on these terms."* \
> **We are revising the paper to better clarify and improve the presentation.**
>
> * *"The novelty is limited"* \
> *"Novelty is incremental: the authors only combine existing quantization techniques with MoE layers. There is no new technique proposed except a quantization recipe."* \
> **As we mention in the general response, we are focusing on the MoE layers' unique property and extensive analysis on it. Based on that, we propose a simple yet effective way to solve one of the hardest problems to practically deploy MoE models in real-world application. Even though it's simple, we believe this would benefit many researchers in the field. Furthermore, the observations of emerging sparsity could guide us to a better design of MoE models.**
>
> * *"Some statements are not rigorous or lack explanation: for example, using jittering noise and balancing loss cannot uniformly distribute expert utilization (or at least need to show the patterns);"* \
> **Thank you for the comment. We will clarify those in the revision.**
>
> * *"it is not clear why symmetric quantization can give an advantage to quantize many weights near zero, since as long as the ''0'' bin covers many weights."* \
> **If the weights are normally distributed, many of the weight values are concentrated near the mean value. With symmetric quantization, those mean values will always map to zero. On the other hand, with non-symmetric quantization, the mean values can map to non-zero values when the absolute values of maximum and minimum are different. This is if the peak in the normal distribution is mapped to zero or not when the values are quantized. We will try to provide some more clarification in the paper.**

---

### Official Review · Reviewer_5W77 · 2022-10-30

**Confidence:** 4
**Correctness:** 4
**Technical Novelty And Significance:** 1
**Empirical Novelty And Significance:** 2
**Recommendation:** 3

**Clarity, Quality, Novelty And Reproducibility:**

The authors make a great effort at describing their experimental setup and I believe that all required details are reasonably disclosed. However, I would like to point out some writing issues that should be addressed:
- Generally, whenever there is times X smaller/larger, these sentences are very confusing. Examples: "we show the model size can be reduced 4.9X smaller than" or "This cuts down the model size of 5.3B parameters from 8.4x of the dense model to only 1.7x dense model".
- other unclear sentences: "the actual inference speed decreases slower than 40% of ", "weights normally distributed centered around zero", "we use the second optimal value algorithm which results in better quantization accuracy". For instance, for the last sentence: was there a first optimal value algorithm? (answer: no) What is quantization accuracy? (should be accuracy of quantized model)



**Strength And Weaknesses:**

After reading the paper I can identify following strengths and weaknesses. Strengths first:
- The paper is one of the first to give an empirical study of quantization of MoE networks. It would be a good manual/starting point for practitioners in the field.

Weaknessess:
- Thoroughness: Despite having good results and having investigated several quantization options, one would still have questions of "what if?" style. There are many additional experiments and empirical evaluations that are needed to make it a stronger contribution, and to be certain of presented recommendations. For instance here are additional questions: 1) if inference happens in fp16, why to stick with uniform or log-uniform quantization schemes? how about non-inform quantization akin k-means? 2) why not to consider finer grouping for quantization instead of per-tensor and per-channel? 3) why PTQ calibration techniques are not discussed? are all calibrations work the same? 4) what is the tradeoff between # experts vs bit-width of compression? are there certain recommendation? and many other questions of this format
- The paper would benefit from another proof-reading pass: there are many places where it is hard to understand what was exactly meant.


**Summary Of The Paper:**

The paper studies the effects and technique of quantizing the mixture of experts (MoE) models originating in the field of NLP. These MoE models are constructed to have gating layers that route inputs through certain matrices, and typically, the chosen gating settings are such that overall FLOPs are the same as of a single (non-MoE) network. As such, the only issue is saving and loading the larger MoE network: according to Table 1, MoE network of 32 experts is requires 9x more storage and has only 0.4x thorouput of non MoE network (despite having same theoretical FLOPs). In this paper, authors empirically study whether quantization is a viable mechanism to reduce the size of MoE networks and present their findings: using 2-bit quantization on expert layers and performing additional finetuning (QAT), it is possible to achieve an MoE network of 32 experts that is only 1.7x times larger than dense baseline (non-compresses MoE is 8.4x of dense network) Their full recipe is available in Section 3.3.

**Summary Of The Review:**

Overall, the main contribution of the paper is empirical evaluation of various quantizaiton options/techniques when applied to MoE networks. Although I believe such evaluations would be very helpful to the community, the current state of this paper requires having additional empirical studies to make its contribution/claims stronger.

---

> ### Author Response · Authors · 2022-11-17
> **Response to Reviewer 5W77**
>
> Thank you so much for the constructive comments.
>
> * *"The paper is one of the first to give an empirical study of quantization of MoE networks. It would be a good manual/starting point for practitioners in the field."* \
> **Thanks for the comment. Yes, we hope our work provides a good starting point for a better MoE model design.**
>
> * *"Thoroughness: Despite having good results and having investigated several quantization options, one would still have questions of "what if?" style. There are many additional experiments and empirical evaluations that are needed to make it a stronger contribution, and to be certain of presented recommendations. For instance here are additional questions: 1) if inference happens in fp16, why to stick with uniform or log-uniform quantization schemes? how about non-inform quantization akin k-means? 2) why not to consider finer grouping for quantization instead of per-tensor and per-channel? 3) why PTQ calibration techniques are not discussed? are all calibrations work the same? 4) what is the tradeoff between # experts vs bit-width of compression? are there certain recommendation? and many other questions of this format"* \
> **As we mention in the general response (1 & 2), we are focusing on the MoE layers' unique property and extensive analysis on it. Based on that, we propose a simple yet effective way to solve one of the hardest problems to practically deploy MoE models in real-world application. Because the simple quantization could preserve the most accuracy, we believe that we may not need to employ more complicated methods. Even though it's simple, we believe this would benefit many researchers in the field. Furthermore, the observations of emerging sparsity and robustness to the perturbation could guide us to a better design of MoE models.** \
> **Also, as we mention in the general response 3, we actually conducted extensive experiments including a different task, different model sizes, and data. Due to the page limitation, we couldn't include them in the main body of the paper. We will clarify this fact in the paper.**
>
> * *"The paper would benefit from another proof-reading pass: there are many places where it is hard to understand what was exactly meant."* \
> *"Generally, whenever there is times X smaller/larger, these sentences are very confusing. Examples: "we show the model size can be reduced 4.9X smaller than" or "This cuts down the model size of 5.3B parameters from 8.4x of the dense model to only 1.7x dense model"."* \
> *"other unclear sentences: "the actual inference speed decreases slower than 40% of ", "weights normally distributed centered around zero", "we use the second optimal value algorithm which results in better quantization accuracy". For instance, for the last sentence: was there a first optimal value algorithm? (answer: no) What is quantization accuracy? (should be accuracy of quantized model)"* \
> **Thank you for the comments. We will clarify those in the revision.**

---

### Official Review · Reviewer_MzdQ · 2022-11-03

**Confidence:** 4
**Correctness:** 3
**Technical Novelty And Significance:** 2
**Empirical Novelty And Significance:** 2
**Recommendation:** 5

**Clarity, Quality, Novelty And Reproducibility:**

This paper clearly explains the challenges for MoE deployment and introduce its weight-only technique. This paper needs more valuable analysis instead of the common phenomenon in quantization.

**Strength And Weaknesses:**

Strength: The paper applies the quantization to reduce the model size and accelerate the model inference. The paper conduct several experiments to show the effectiveness of the method.

Weakness: This paper lacks the comparison with other quantization methods for MoE. This paper needs more valuable analysis for applying the quantization into MoE since using quantization to reduce the model size to speedup is a common technique and result.

**Summary Of The Paper:**

This paper proposes a simple weight-only quantization method to quantize the MoE model. This method combines several common quantization techniques to quantize and reduce the model size. The experiment results show that the method can be used for accelerating the MoE model.

**Summary Of The Review:**

This paper proposes a simple weight-only quantization method to quantize the MoE model. The experiment results show that the method can be used for accelerating the MoE model. However, this paper lacks a comparison with other quantization methods for MoE, and there is not any new technique for solving the MoE. Moreover, this paper needs more polishes. To sum up, I do not recommend this paper.

---

> ### Author Response · Authors · 2022-11-17
> **Response to Reviewer MzdQ**
>
> Thank you so much for the constructive comments.
>
> * *"This paper lacks the comparison with other quantization methods for MoE. This paper needs more valuable analysis for applying the quantization into MoE since using quantization to reduce the model size to speedup is a common technique and result."*
>
> **As we mention in the general response (1 & 2), we are focusing on the MoE layers' unique property and extensive analysis on it. Based on that, we propose a simple yet effective way to solve one of the hardest problems to practically deploy MoE models in real-world application. Because the simple quantization could preserve the most accuracy, we believe that we may not need to employ more complicated methods. Even though it's simple, we believe this would benefit many researchers in the field. Furthermore, the observations of emerging sparsity and robustness to the perturbation could guide us to a better design of MoE models.**
>
> **Also, as we mention in the general response 3, we actually conducted extensive experiments including a different task, different model sizes, and data. Due to the page limitation, we couldn't include them in the main body of the paper. We will clarify this fact in the paper.**

---

### Author Response · Authors · 2022-11-17
**General response to all reviewers**

We thank all reviewers for the constructive and insightful comments. We would like to clarify some of the points the reviewers commented on.

1. **The main motivation and the goal of the paper:** \
In our initial investigation, we found that MoE layers behaved very differently than the conventional neural network layers in terms of robustness to the perturbation. This hinted us that simple methods could significantly improve the usability of the MoE models in the real-world. So far, even though MoE models provide good quality, the practical usage was limited by the throughput/speed and size challenges.

2. **Comparison to the more sophisticated quantization methods:** \
We believe the upper bound of the quantized or approximated models would be the original models’ quality without approximation. If the approximation methods can provide close enough quality to the original model, we think we may not need to make the solution more complicated. Rather, we chose to analyze how MoE layers and conventional NN layers behave differently on the same method. Also, we investigated why those phenomena are happening by looking at the distribution of model weights.

3. **Extensive experiments in the Appendix:** \
To understand if the phenomena are specific to a model / a task or not, we conducted various experiments with a different task, model sizes, data, etc. However, due to the page limit, we could only include the representative model case in the main paper. As a result, in the appendix, we provide various experiments with a different task (abstractive summarization), different model sizes, data sizes. We observed that our findings could be seen in all those different cases.

---

### Decision · Program_Chairs · 2023-01-20

**Decision:**

Reject

**Justification For Why Not Higher Score:**

Purely empirical benchmarking without offering much of new insights

**Justification For Why Not Lower Score:**

N/A

**Metareview: Summary, Strengths And Weaknesses:**

The paper presents the first empirical study of quantization on MoE networks. The topic makes a good practitioner's study that the field must look into anyway.

Although the observations are interesting, there is little technical innovation based on them. The paper primarily reports an observational study that applies known quantization techniques to known architectures. It is unclear what unique challenges the authors tackled. Also, to make the empirical benchmarking more valuable, the authors need to have more comprehensive studies: several suggested by Reviewer 5W77 are good points.

Unfortunately, the reviewers are almost unanimously negative regarding this paper. The authors provided rebuttal, but tend to be brief and many answers are hand-waving still. AC reads all comments and the paper, and also feels the current shape cannot reach the ICLR bar.